# Functional Adaptation of Vocalization Revealed by Morphological and Histochemical Characteristics of Sonic Muscles in Blackmouth Croaker (*Atrobucca nibe*)

**DOI:** 10.3390/biology11030438

**Published:** 2022-03-13

**Authors:** Hung-Tai Lee, Bao-Quey Huang, Cheng-Hsin Liao

**Affiliations:** 1Department of Environmental Biology and Fisheries Science, National Taiwan Ocean University, Keelung 20224, Taiwan; htlee@ntou.edu.tw (H.-T.L.); b0039@mail.ntou.edu.tw (B.-Q.H.); 2Center of Excellence for the Oceans, National Taiwan Ocean University, Keelung 20224, Taiwan

**Keywords:** sciaenid, bioacoustics, fiber type, morphology, myosin adenosine triphosphatase, extrinsic

## Abstract

**Simple Summary:**

Sound production is common in numerous fish species. Some species can emit calls through the contraction of specialized muscles called sonic or drumming muscles. The sonic muscles of fish are among the fastest muscles in vertebrates. Although numerous studies have investigated the mechanism underlying sound production in fish, only the distinct features of the sonic muscles of a few species have been investigated. We demonstrated that the sonic muscles have functionally adapted for fast twitching and fatigue resistance, which support vocalization in the blackmouth croaker (*Atrobucca nibe*).

**Abstract:**

Sound production in the blackmouth croaker (*Atrobucca nibe*) was characterized using acoustic, morphological, and histochemical methods. Their calls consisted of a train of two to seven pulses; the frequency ranged from 180 to 3000 Hz, with a dominant frequency of 326 ± 40 Hz. The duration of each call ranged from 80 to 360 ms. Male *A. nibe* possess a pair of bilaterally symmetric sonic muscles attached to the body wall adjacent to the swim bladder. The average diameter of the sonic muscle fibers was significantly shorter than that of the abdominal muscle fibers. Semithin sections of the sonic muscle fibers revealed a core-like structure (central core) and the radial arrangement of the sarcoplasmic reticulum and myofibrils. Numerous mitochondria were distributed within the central core and around the periphery of the fibers. Most of the fibers were identified as Type IIa on the basis of their myosin adenosine triphosphatase activities, but a few were identified as Type IIc fibers. All sonic muscle fibers exhibited strong oxidative enzyme activity and oxidative and anaerobic capabilities. The features suggest that the sonic muscles of *A. nibe* are morphologically and physiologically adapted for fast twitching and fatigue resistance, which support fish vocalization.

## 1. Introduction

Animal communication is the transmission of information between individuals through various sensory signals, such as light, sound, chemicals, touch, hydrodynamics, and electricity [1]. For marine organisms, sound signals are the most effective sensory modality because, in aquatic environments, they travel faster and farther than any other sensory signals [2,3]. Sound production has been widely observed in teleost species [4,5] and is correlated with behavioral responses such as feeding, aggression, territorialism, and reproduction [6]. The mechanisms of sound production in fish are complex and diverse, with three major mechanisms being involved [4,7]. First, hydrodynamic sounds are produced through changes in direction and velocity as fish move. Second, stridulatory sounds are produced by friction of the teeth, fin spines, or bones of fish. Third, swim bladder sounds are produced by the contraction of specialized muscles known as sonic or drumming muscles. The second and third mechanisms are mainly involved in the vibration of the swim bladder, particularly in the case of sonic muscles [4,8]. 

Sonic muscles are specialized muscles used for sound production in fish species and can be intrinsic or extrinsic depending on their origin and attachment to the swim bladder [4,9]. Intrinsic sonic muscles are exclusively attached to the swim bladder wall in fish species such as *Opsanus tau*, *Prionotus carolinus*, and *Zeus faber* [10,11,12]. Extrinsic sonic muscles typically originate either on the cranium, pectoral girdle, or lateral body wall musculature and are inserted into the swim bladder or adjacent tendons and bones in fish species such as *Cynoscion regalis*, *Hemitaurichthys polylepis*, and *Terapon jarbua* [13,14,15]. Intrinsic and extrinsic sonic muscles have both been observed in species such as cowfish and boxfish [16,17]. 

The sonic muscles of fish are among the fastest muscles in vertebrates [18,19,20]. However, they are not necessarily fast-twitch muscles because slow-twitch sonic muscles have also been reported [16,17,21]. The acoustic characteristics of soniferous fishes have been well-documented; by contrast, the functional characteristics of sonic muscles have rarely been investigated, and studies have been limited to a few soniferous fishes [4]. The morphological and histochemical characteristics of skeletal muscles have been associated with their function and metabolic activity. For example, the myosin adenosine triphosphatase (mATPase) activity of muscle fibers is closely related to their contraction rate [22,23,24]. The oxidative enzyme activities of muscle fibers, namely nicotinamide adenine dinucleotide dehydrogenase-tetrazolium reductase (NADH-TR) and lactate dehydrogenase (LDH), are often used as indicators of oxidative capacity and anaerobic metabolism, respectively [24].

Sciaenidae are one of the largest families of soniferous fishes, comprising approximately 283 species within 67 genera [25]. Sciaenid species are well-known to produce sounds primarily for courtship during the spawning season [26,27,28]. A member of Sciaenidae, *Atrobucca nibe*, also known as “blackmouth croaker”, is widely distributed in the Indo-West Pacific. The species is often found in coastal waters with a depth of approximately 45-200 m and tends to inhabit seaweed beds and gravel [29]. As *A. nibe* is generally considered a commercial species, investigations into it rarely consider all relevant aspects. Furthermore, many offshore wind farms are planned to be constructed in the coastal water off Taiwan, and their impacts on fish species have also raised public awareness. However, our understanding of this commercially important fish species is still limited.

In this study, the calls emitted from *A. nibe* were recorded and analyzed. We further investigated the mechanism underlying sound production in this species by using morphological and histochemical methods. Morphological and histochemical examinations of the sonic muscles were conducted to determine the distinct physiological functions and metabolic activities related to sound production in *A. nibe*. The findings provide a comprehensive understanding of the morphological and physiological adaptation supporting the vocalization in *A. nibe* and its implication for muscle biology. Moreover, this study could also offer the fundamental knowledge required for monitoring this species in the wild.

## 2. Materials and Methods

### 2.1. Specimen Collection

A total of 349 (152 female and 197 male) *A. nibe* caught by long lines and trawl fisheries in the coastal waters off southwestern Taiwan were sampled. The sex, total length, standard length (SL), total weight, and gutted weight of each fish were recorded. Ten individuals (SL: 22–26 cm) captured by hook and line in the spawning season (April) were used for sound recording. The hooks were gently removed from the fish to minimize stress and increase survival time. The live specimens were individually placed in a 150-L polystyrene tank to record their calls. 

### 2.2. Sound Recording and Analysis

The acoustic signals emitted by individual fish were detected using an underwater microphone (Hydrophone Type 8104, frequency range: 0.1–80 kHz, Brüel & Kjær, Naerum, Denmark), amplified by a preamplifier (NEXUS conditioning amplifiers, model 2690, Brüel & Kjær, Naerum, Denmark), and recorded with a digital recorder (minidisk recorder XM-D1, JVC, Yokohama, Japan). The acoustic signals were analyzed using Avisoft-SASLab Pro (Avisoft Bioacoustics, Berlin, Germany) at a sampling rate of 16 kHz and a resolution of 16 bits [30]. The number of pulses per call, pulse period (the time between one pulse and the next), and the call duration (measured from the beginning of the first pulse to the beginning of the last pulse) per call were quantified using sonograms [27]. The dominant frequency of a call was determined by isolating each pulse from multi-pulse calls, and its power spectrum was derived through a fast Fourier transform. The highest peak in the power spectrum was defined as the dominant frequency.

### 2.3. Morphometry of Sonic Muscles

To determine the position of the sonic muscles and their relationships with other structures, a cross section was made at approximately 50% SL. The weight, width (anterior–posterior axis of the muscle), length (dorsoventral axis of the muscle), and maximal thickness (cross section of the muscle) of the sonic muscles were measured. 

### 2.4. Histology

The sonic muscles along the width in the mid region were dissected (5 × 5 mm^2^) and promptly immersed in isopentane cooled by liquid nitrogen. Serial 10-μm transverse sections of muscle blocks were taken using a cryostat (BRIGHT OTF/AS-001, Cambridge, UK), maintained at −20 °C, and stained with haematoxylin and eosin to visualize the histological characteristics of the sonic muscle fibers. Skeletal muscles in the lateral body wall adjacent to the sonic muscles were also sampled and prepared for comparison with the sonic muscles. To further examine the structure of the sonic muscles, sample specimens were fixed in 3% glutaraldehyde and 3.84% paraformaldehyde in 0.2 M phosphate buffer at pH 7.4 for 1 h, washed in phosphate buffer, and postfixed in 1% osmium tetroxide for 3 h. Subsequently, the samples were repeatedly washed with phosphate buffer, dehydrated through a series of graded ethanol and propylene oxide baths, and embedded in Epon resin. Semithin sections (600 nm) were cut on an ultramicrotome (RMC, MTX, Tucson, AZ, USA) and stained with toluidine blue.

### 2.5. Histochemistry

To demonstrate the histochemical characteristics of the sonic muscles, serial cross sections of the sonic muscle fibers were preincubated in a pH 9.4–9.9 (in 0.1-pH increments) alkaline preincubation solution for 15 min or a pH 4.9–5.4 (in 0.1-pH increments) acidic preincubation solution for 5 min at room temperature before the calcium method was applied for the adenosine triphosphatase (ATPase) response to differentiate the fiber types [24]. The oxidative enzyme activities of NADH-TR and LDH in the muscle fibers were examined using a previously described method [13]. All slides were examined and photographed with an Olympus photomicroscope BX50 (Olympus, Tokyo, Japan). The cross-sectional area of the muscle fibers was measured using an image analysis system (Measurement Tools, CHEMAX International Corporation, Taipei, Taiwan), and the diameter of the muscle fibers was subsequently calculated.

### 2.6. Statistics

Statistical analysis was performed in SPSS 19 (IBM, Armonk, NY, USA), and the results are presented in terms of the mean ± standard deviation. All data were verified to be normally distributed through the Shapiro–Wilk test. The data were subjected to Student’s *t* test to examine differences in the length, width, thickness, and weight of both sides of the sonic muscles. 

## 3. Results

### 3.1. Sound Characteristics

The calls of *A. nibe* mainly consisted of a series of two to seven pulses at a regular pulse period of 52 ± 9 ms (Figure 1A,E). Each pulse consisted of 2–3 cycles of acoustic energy (Figure 1B). The frequency of their calls ranged from 180 to 3000 Hz (Figure 1C), with a dominant frequency of 326 ± 40 Hz (Figure 1D). The duration of each call mainly ranged from 80 to 360 ms (Figure 1F).

### 3.2. Structure and Organization of Sonic Muscles

In the *A. nibe* specimens, bilaterally paired sonic muscles were attached to the body wall adjacent to the swim bladder and were nearly the entire length of the body cavity (Figure 2A,B). The sonic muscle fibers originated from the abdominal hypaxial muscles, ran vertically against the axis of the body, and were inserted into a central tendon (aponeurosis) over the dorsal surface of the swim bladder (Figure 2B,C). The sonic muscles were closely associated with but not directly attached to the swim bladder. The surface of the sonic muscles was covered by a peritoneal sheet ventrally. The peritoneum was attached to the ventrolateral surface of the swim bladder (Figure 2B). The swim bladder was thus retroperitoneal in position, and the sonic muscles were inside the peritoneum (Figure 2B). The dorsal side of the sonic muscles (about one-quarter the length of the sonic muscle) was tightly bound to the abdominal body wall musculature by connective tissues (Figure 2B,C); this was typical of the extrinsic sonic muscles. The color of the fresh sonic muscles was red (Figure 2D). 

The extrinsic sonic muscles were only identified in the males and completely absent in all females examined in this study. The sonic muscles were also absent in the males shorter than 120 mm in SL, in which the gonads were immature and indistinguishable. No significant differences were observed in length, width, thickness, and weight between the left and right sides of sonic muscles (*t* test, *p* > 0.05, *n* = 179). However, the length, width, thickness, and weight of the sonic muscles increased with the SL of the fish (Figure 3).

### 3.3. Morphological Characteristics of Sonic Muscles

The sonic muscle fibers had a pale central core surrounded by a dark region containing the myofibrils (Figure 4A). The diameter of the sonic muscle fibers (40 ± 4 μm, *n* = 100) was smaller than that of the body wall muscle in the abdominal cavity (110 ± 9 μm, *n* = 100) (Figure 4A,B). The semithin cross sections of the sonic muscle fibers revealed a notable central core (Figure 4C). The myofibril and sarcoplasmic reticulum were closely tight as alternating ribbons that radially extended from the central core to the periphery of the fibers. A considerable number of mitochondria were distributed within the internal central core and periphery of the sonic muscle fibers (Figure 4D).

### 3.4. Histochemical Characteristics of Sonic Muscles

The sonic muscles exhibited diversity in mATPase activity (Figure 5A). The serial sections of the sonic muscle fibers revealed that most of the sonic muscle fibers exhibited positive activity (Figure 5B) under alkaline conditions (pH 9.7) (Figure 5B) and negative activity (Figure 5C,D) under acidic conditions (pH 4.9 and pH 5.1). The sonic muscle fibers were classified as Type IIa or IIc on the basis of their mATPase activity under each condition (Table 1). Several muscle fibers (<1%) exhibited positive activity under alkaline (pH 9.7) and acidic conditions (pH 5.1 and pH 5.3; Figure 5C,D). These sonic muscle fibers were classified as Type IIc (Table 1). As for the oxidative enzyme activities (NADH-TR and LDH), the sonic muscle fibers exhibited positive NADH-TR and LDH activities, especially at the central core and periphery (Figure 5E,F). The diameter of the Type IIa fibers (43 ± 4 μm, *n* = 200) was significantly larger than that of the Type IIc fibers (29 ± 4 μm, *n* = 80; Table 1).

## 4. Discussion

Although sound production in sciaenid species has been observed by fisherman and researchers, the characteristics of their sounds have only been documented for 24 species, less than 10% of the *Sciaenidae* family [28]. In general, sciaenid sounds are low-frequency calls with numerous pulses and a fundamental frequency below 1 kHz, and differences in sounds between species have been observed [28,31,32]. In this study, the calls emitted from *A. nibe* were characterized by a distinct dominant frequency of 326 ± 40 Hz, two to seven pulses, and a duration of 80–360 ms. These features could be species specific for *A. nibe*. However, sound characteristics also vary depending on sex, body size, and environmental factors [14,27,31,33]. Hence, subsequent studies should investigate the sound characteristics of all sizes of *A. nibe* under various environmental conditions. Nevertheless, the dominant frequency, pulse number, and duration of the *A. nibe* calls observed in this study can serve as reference for the passive acoustic monitoring of soniferous fish species [34,35], fishery management, and conservation of the species.

Fish employ various mechanisms to produce sound. Swim bladder sounds are produced through the contraction of specialized sonic muscles [4]. The sonic muscles in *A. nibe* suggest that the fish use this mechanism to produce sound. The sonic muscles are extrinsic in *A. nibe* because they do not directly connect to the swim bladder.

We observed that sonic muscles were absent in *A. nibe* shorter than 120 mm in SL, at which size their gonads are either immature or indistinguishable. Sonic muscles are only present in male *A. nibe* and can be considered a secondary sex characteristic. In most sciaenid species, sonic muscles are present only in males [36,37,38,39], whereas they are present in both sexes of sciaenid species such as *Argyrosomus japonicus*, *Micropogonias undulatus*, and *Pogonias cromis* [27,36,40]. The length, width, thickness, and weight of the sonic muscles generally increase with fish length for *A. nibe*; the low *R*^2^ in the present linear regression analysis might be attributable to seasonal variation in their sonic muscles (unpublished data), as reported in other sciaenid species [26,41].

The distinct characteristics of the sonic muscles are closely related to their function. We observed several unique morphological and histochemical characteristics of sonic muscles that support vocalization in *A. nibe*. The diameter of the sonic muscle fibers was significantly smaller than that of the abdominal muscles in *A. nibe*. Similar findings have been reported in both related species (*A. regius*, *C. regalis*, *Sciaenops ocellatus*) [28,42,43,44] and non-related species (*Carapus acus*, *L. cornuta*, *O. tau*) [17,45,46]. Due to their small diameter, sonic muscle fibers have a high surface-to-volume ratio, which facilitates the transportation and diffusion of nutrients (glucose and oxygen) and metabolic products (lactate and carbon dioxide) [17,44,45]. Consequently, the contraction of the sonic muscles for sound production can benefit from efficient energy supply and the removal of metabolic waste.

The radial arrangement of the sarcoplasmic reticulum and myofibrils in *A. nibe* could shorten the travel distance of calcium ions between contractile protein and the sarcoplasmic reticulum, resulting in effective calcium ion transportation. In addition, the numerous mitochondria around the central core and periphery of the sonic muscle fibers provide energy for muscle contraction and improve the sustained contraction of the sonic muscles, as reported in other studies [44,47,48].

The ATPase activity of the muscle fibers suggests a relationship with contraction rate [24]. We observed that the majority of the sonic muscle fibers were functionally equivalent to Type IIa fibers in mammals with alkali-stable ATPase activity [24]. This finding also suggests that the sonic muscle fibers of *A. nibe* are fast oxidative glycolytic and fatigue resistant [24]. Similar studies have been conducted on three soniferous species, namely *C. acus*, *O. tau*, and *T. jarbua* [13,46,49], but this is the first study on the sciaenid species. We observed several sonic muscle fibers functionally equivalent to Type IIc fibers in mammals with acid- and alkali-stable ATPase activity [24]. Type IIc fibers are considered precursor cells and have been observed in fish muscles with newly formed muscle fibers [50], a process called hyperplasia. In *O. tau*, sonic muscles consist of Type IIa fibers and lack Type IIc fibers [49]. Type IIc fibers have been observed in *T. jarbua* [13] and *A. nibe* (this study). Additional studies are required to elucidate the role of Type IIc fibers in sonic muscles.

Strong NADH-TR and LDH activities were observed in the sonic muscle fibers of *A. nibe*. The high NADH-TR activity indicates that the sonic muscle fibers can obtain ATP through the electron transport system and provide energy required for fast-twitch action [13,24,49]. The high LDH reaction activity suggests that the sonic muscle fibers use pyruvate for anaerobic metabolic processes [13,24].

## 5. Conclusions

This study investigated the mechanism underlying sound production in *A. nibe* through a comprehensive method. The calls emitted by *A. nibe* had distinct sound parameters, namely dominant frequency, pulse number, and call duration. Extrinsic sonic muscles are bilaterally symmetric and present only in male *A. nibe*. Due to their small diameter and distinct structure (central core and radial arrangement of the sarcoplasmic reticulum and myofibrils), the sonic muscles in *A. nibe* benefit from efficient energy supply and the removal of metabolic waste. The histochemical characteristics of the sonic muscle fibers indicated rapid contraction and strong oxidative and anaerobic capabilities. These features suggest that the sonic muscles of *A. nibe* are morphologically and physiologically adapted for fast twitching and fatigue resistance, supporting fish vocalization. While this is the first study to fully investigate sound production and sonic muscles in *A. nibe*, major findings from this work are generally agreed with the previous studies in other soniferous species. Perhaps the most striking finding is the identification of two distinct fiber types in sonic muscles of *A. nibe*. The sonic muscles mainly consisted of Type IIa fibers, but a few Type IIc fibers were observed in *A. nibe*. Type IIc fibers are considered as the indicator for the hyperplastic growth in fish muscle. The exact function of the Type IIc fibers in the sonic muscles remains to be investigated. The sonic muscles of soniferous fish are complex and diverse. Comprehensive studies on sonic muscles can provide novel insights into muscle biology.

## Figures and Tables

**Figure 1 biology-11-00438-f001:**
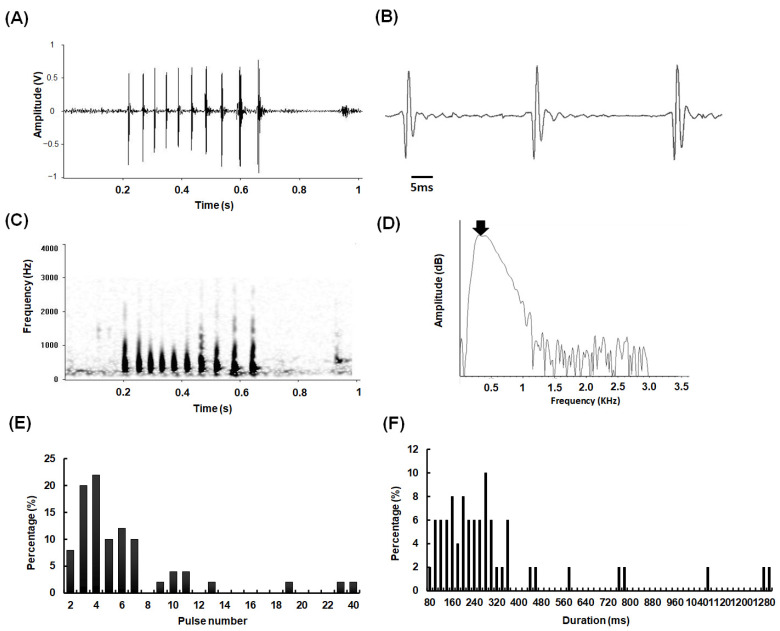
Acoustic characteristics of sounds emitted from *A. nibe*: (**A**) oscillogram, (**B**) expanded oscillograms, (**C**) sonogram, (**D**) and power spectrum of a single pulse. (**E**) Histogram of pulse number and (**F**) call duration. The arrow indicated the dominant frequency with the maximal energy. A total of 50 calls emitted from 10 individuals (SL: 22–26 cm) were analyzed.

**Figure 2 biology-11-00438-f002:**
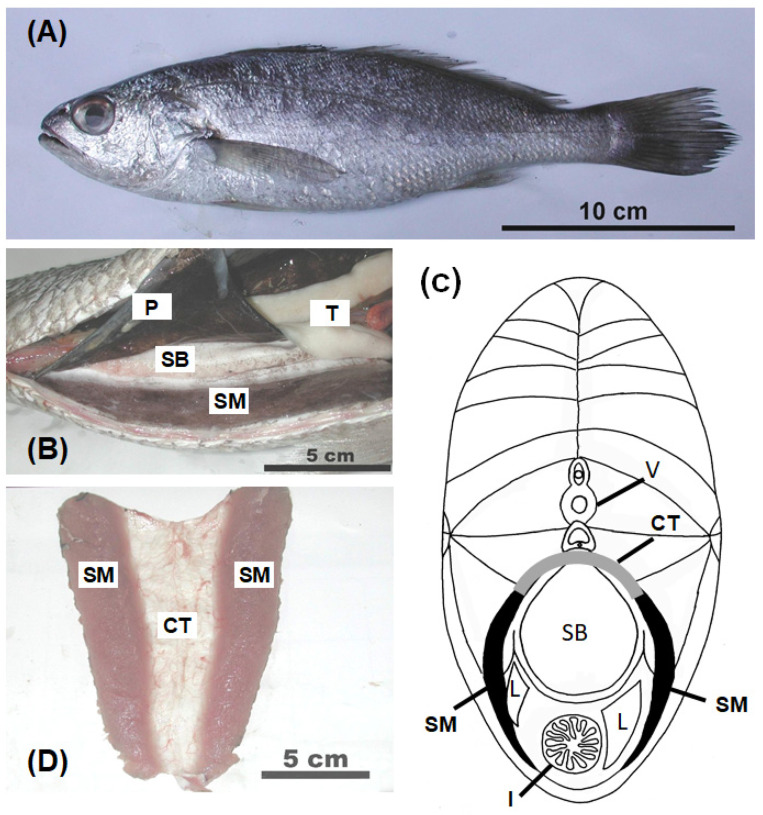
Structure and organization of sonic muscle in *A. nibe*. (**A**) Lateral view of *A. nibe*. (**B**) Ventral view of extrinsic sonic muscle attached to lateral body wall (hypaxial muscle). Extrinsic sonic muscles attached to body wall and separated from the swim bladder by a peritoneal sheet. (**C**) Transverse view of *A. nibe* at 50% SL. (**D**) Bilaterally paired sonic muscles are connected by a central tendon (aponeurosis). CT: central tendon; I: intestine; L: liver; P: peritoneum; SM: sonic muscle; SB: swim bladder; V: vertebra.

**Figure 3 biology-11-00438-f003:**
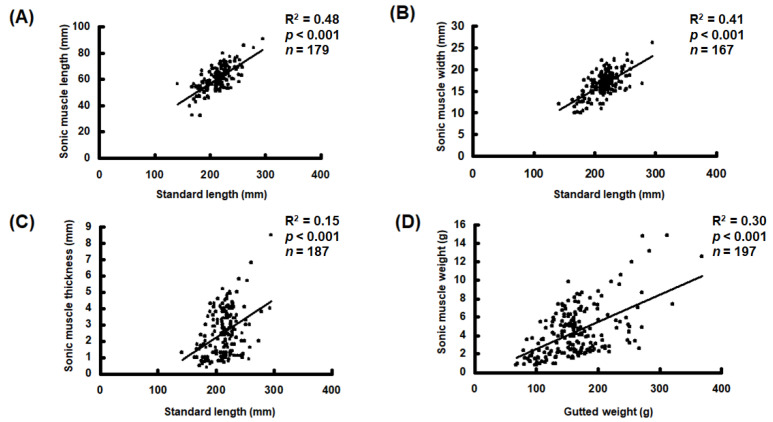
Morphometric analysis of sonic muscles in *A. nibe*. Linear regression of length (**A**), width (**B**), thickness (**C**), and weight (**D**) of sonic muscles against fish size.

**Figure 4 biology-11-00438-f004:**
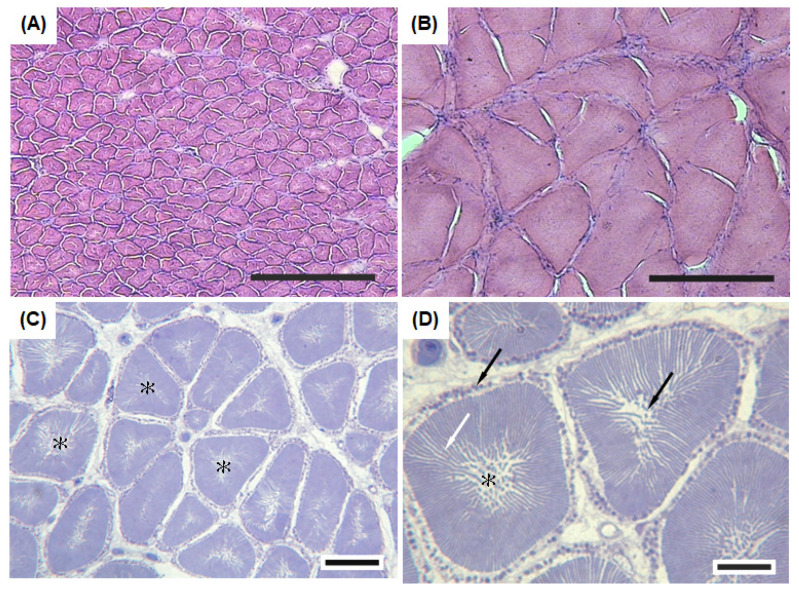
Histological characteristics of sonic muscle fibers in *A. nibe*. Frozen cross section of (**A**) sonic muscle fibers and (**B**) their adjacent hypaxial muscle fibers stained with haematoxylin and eosin. The cross-sectional area of the sonic muscle fibers was smaller than that of the hypaxial muscle fibers. The semithin cross sections of the sonic muscle fibers indicated (**C**) consistent fragmented core-like structures (asterisk) in the central region. (**D**) High magnification of the cross-sectional areas of the sonic muscle fibers revealed alternating dark and light bands (sarcoplasmic reticulum and myofibrils) within the contractile cylinder (white arrow), and numerous mitochondria were distributed within the fragmented central core and at the periphery of the contractile cylinder (black arrow). Scale bar = 200 μm (**A**,**B**); 25 μm (**C**); 10 μm (**D**); SL = 26 cm.

**Figure 5 biology-11-00438-f005:**
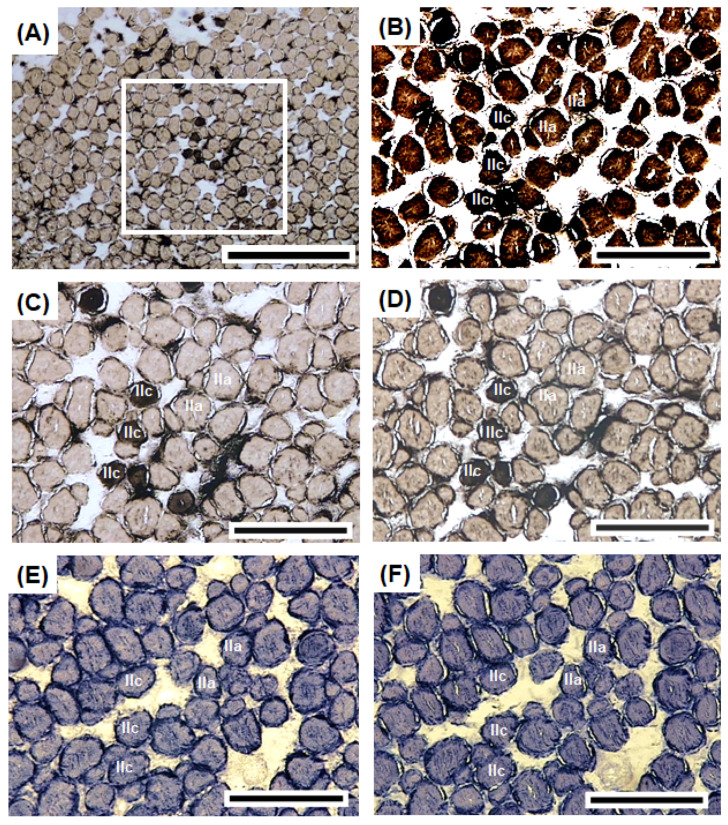
Histochemical characteristics of sonic muscle fibers in *A. nibe*. (**A**) mATPase activity of sonic muscle fibers. Several fibers with strong ATPase activity in the sonic muscles preincubated at pH 4.9 exhibited nonhomogeneous ATPase activity under lower magnification (**A**). IIa fibers exhibiting strong ATPase activity in the sonic muscles preincubated under alkaline conditions (pH 9.7) (**B**). IIa fibers exhibiting weak ATPase activity in the sonic muscles preincubated under acidic conditions (pH 4.9, 5.1) (**C**,**D**). IIc fibers exhibiting strong ATP activity in the sonic muscles preincubated under alkaline conditions (pH 9.7) and (**B**) weak ATPase activity in the sonic muscles preincubated under acidic conditions (pH 4.9, 5.1) (**C**,**D**). The Type IIa and IIc fibers exhibited strong NADH-TR and LDH activity (**E**,**F**). Scale bar = 500 μm in (**A**) and 200 μm in (**B**–**F**).

**Table 1 biology-11-00438-t001:** Histochemical characteristics of Type IIa and IIc fibers in the sonic muscles of *A. nibe*.

	Type IIa	Type IIc
ATPase (pH 9.7)	+++	++
ATPase (pH 5.1)	−	+++
ATPase (pH 4.9)	−	+++
NADH-TR	+++	+++
LDH	+++	+++
Composition (%)	>95	
Diameter (μm) *	43 ± 4 (*n* = 200)	29 ± 4 (*n* = 80)

Staining intensity: − = no staining; ++ = moderate staining; +++ = heavy staining. *: Combined data from 6 indivduals (SL: 25 ± 1 cm).

## Data Availability

The data presented in this study are available on request from the corresponding author.

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
