# Peer review of "Functional Adaptation of Vocalization Revealed by Morphological and Histochemical Characteristics of Sonic Muscles in Blackmouth Croaker (Atrobucca nibe)"

_biology, 2022, doi:10.3390/biology11030438_

Round 1

Reviewer 1 Report

Lee et al's presentation is largely cleaned up and makes a nice contribution to fish bioacoustics and specialized muscles.

One thing that still needs clarification is the sample size in statistics comparing sonic and trunk muscle fibers (L 141, 184 and 210). The authors need to provide additional information on the number of fish sampled. If they used one fish as representative that would be ok, but they need to be explicit: They measured 100 sonic fibers and 100 trunk fibers. If they used one fish, then they should not use t tests and just report the results (mean and SD). If they used multiple fish, how this was done should be specified.

L 189. Substitute mitochondria for particles.

Author Response

Dear Reviewer,

Reviewer 2 Report

The current study describes sounds, the sound generating mechanism and the histochemistry of sonic muscle in a family of bony fishes well-know for their sound production. The common names of the sciaenid family namely croakers or drums indicate this clearly.

Sounds and sonic muscles have been described in numerous species of croakers and current results are in accordance with prior studies on this family and other vocal fish families.

I think authors should try to point out what is new in their study as compared to previous ones. 

Minor comment

Several details are missing in the Methods section.

Abstract

Line 18 + 91+145+128+285: The term “vocal calls” does not make sense; either use “calls”, “vocalizations” or “sounds”

Introduction

Line 42-43. Hydrodynamic sounds such as swimming noise may not be listed here because they are not intentionally produced acoustic signals or vocalization but rather cues, namely unintentional information.

Lines 80-82: Unclear what is meant by “functional adaptation of vocalization”. Does it mean they are morphologically and physiologically adapted to generate sounds/calls/vocalizations? Please rewrite this sentence.

Methods

First sentence: Does the total number of fish caught give any information necessary for this study?

Lines 87-89: Give number of males and females and their sizes (SL).

Sound recording:

Give number of sounds recorded per individual, or per sex or in total. Give water temperature during sound recording.

Do authors have any information how many sounds were produced by each individual?

Results

1st paragraph: Explain in detail how means of pulse interval and dominant frequency have been calculated. How many sounds of how many individuals have been analyzed?

Line 100 etc. Explain in Figure 1A and B how sound duration and pulse interval have been  determined. If pulse interval has been measured from the beginning of a pulse to the beginning of the following pulse then this time needs to be termed “pulse period”.  Interval would be the time from the end of a pulse until the beginning of the following pulse.

Lines 147-148. The duration of sounds ranged up to 360 ms. This sentence is in contradiction to Figure 1F which shows a maximal duration of 1280 ms. Please clarify.

Figure 1D: Add an arrow to indicate the dominant frequency in this power spectrum.

Add variable and units to the Y-axis.

Figure 1A+C: The international abbreviation for seconds is s or ms but not sec.

Figure 2: Explain abbreviations in legend: L, T, and delete Ce, because it is not used in this figure.

Line 189: I suggest to change the sentence and write “mitochondria” instead of “particles” to make it clear what is meant.

Discussion

Line 254: I suggest to be more specific in the discussion. Mention the species and family names not just the citation. Three out of four citations in brackets deal with the family Sciaenidae.

Conclusion

The fine structure of swimbladder drumming muscles is very similar in representatives of all fish families possessing drumming muscles even in non-related families such as croakers, various catfish families, toadfishes, cuskeels, etc. All possess small diameter muscle fibers built up of myofibrils and lots of sarcoplasmatic reticulum in between. They are all adapted for high contraction rates to allow the production of low-frequency sounds. Thus the description in A. nibe is very similar to our current knowledge. Authors should try to discuss differences to prior studies.  

References

All scientific names of species are written strangely in the References (but not in the main text). This needs to be corrected.

Opsanus Tau should be written Opsanus tau

Zeus Faber should be written Zeus faber

Etc.

  1. This is a book: Needs name of publisher and city
  2. This is a book chapter. Needs editors of book, publisher, and city. E.g. see citation 4
  3. This is a book chapter, see above comment.

Author Response

Dear Reviewer,
